# Intelligent Feedback Analysis of Fluid–Solid Coupling of Surrounding Rock of Tunnel in Water-Rich Areas

**Tao Zhan [1], Xinping Guo [2,*], Tengfei Jiang [2] and Annan Jiang [2]**

[1] Nanchang Rail Transit Group Co., Ltd., Nanchang 330013, China
[2] Highway and Bridge Institute, Dalian Maritime University, Dalian 116026, China
* Correspondence: guoxinping113@163.com; Tel.: +86-15542519113

**Abstract:** To realize parameter feedback optimization of tunnel construction in water-rich areas, a feedback analysis method for tunnel parameters under fluid–solid coupling conditions was established based on an intelligent optimization algorithm. Firstly, the numerical calculation model was established and solved using the fluid–solid coupling model. In orthogonal design analysis, the displacement of surrounding rock and pore water pressure distribution in different rock mass parameter combinations were obtained, and the learning samples needed for machine learning were established. The input group was surrounding rock displacement and pore water pressure, and the output was rock mass parameters. Then, the Gaussian process algorithm was used to obtain the nonlinear mapping relationship contained in the learning samples. A differential evolution algorithm was used to optimize the critical parameters involved in this process. Furthermore, according to the established regression model and the measured displacement and pore water pressure in the research area, differential evolution was used again to optimize the rock mass parameters and obtain the parameter feedback analysis results. Finally, the inversion values were compared with the actual measured values, and the reliability of the surrounding rock parameters obtained from the feedback analysis was verified, providing an effective method for obtaining surrounding rock parameters for similar projects.

**Keywords:** tunnel engineering; fluid–solid coupling; intelligent feedback analysis; Gaussian process; difference evolution algorithm; water-rich tunnel

## 1. Introduction

The interaction of water and rock in a tunnel constitutes a complex geological system. Due to the discontinuity and heterogeneity of rock and the singleness and randomness of laboratory tests, it is more difficult to obtain accurate surrounding rock parameters in tunnel design and numerical simulation. The displacement back analysis method based on an intelligent algorithm can better solve the above problems [1–4]. In 1971, Kavanagh et al. [5] proposed a method of back analysis of elastic modulus. Lu et al. [6] back-analyzed the surrounding rock elastic modulus, Poisson's ratio and other stratum parameters through actual deformation monitoring data. In the process of back analysis, the selection of a reasonable, intelligent algorithm is helpful in improving the inversion accuracy and efficiency of parameters.

Artificial neural network (ANN), genetic algorithm (GA) and particle swarm optimization (PSO) have been widely used in parameter back analysis [7–12]. Feng et al. [13] combined ANN and GA to form an evolutionary neural network method to identify surrounding rock parameters. Deng et al. [14] used the BP network and GA to back-analyze the elastic modulus of three different geologies by using slope displacement, which improved the calculation efficiency and overcame the defects of narrow application range and slow convergence speed of traditional optimization algorithms. Zhou [15] constructed a GA-BP intelligent feedback system to predict the parameters of the tunnel-surrounding

rock. Wang et al. [16] realized the back analysis of fluid–solid coupling parameters through the hybrid intelligent algorithm of differential evolution algorithm (DE) and PSO. PSO and GA can effectively solve global optimization problems [17]. However, when using a stochastic global optimization algorithm, it is often necessary to evaluate the fitness of a large number of random solutions many times to determine the better solution. The Gaussian process (GP) algorithm is a machine learning regression method developed in recent years. It is mainly based on the statistical theory under the Bayesian framework, and has strong generalization ability and good adaptability in solving small nonlinear sample and high-dimensional regression problems [18,19]. Sun et al. [20] established a probabilistic back analysis method based on Bayesian theory. The research results provide a basis for the establishment of a probabilistic back analysis method of geotechnical engineering parameters. In order to realize the dynamic uncertainty inverse analysis of rock mass parameters with the construction process, Zhang et al. [21] introduced the multi-output support vector machine method and Bayesian theory into the dynamic uncertainty inverse analysis of rock mass parameters. Tao [22] established the probabilistic back analysis and deformation prediction method of rock and soil parameters based on Bayesian theory. Sun [23] studied the geotechnical engineering back analysis method based on multi-objective optimization and Bayesian theory. DE is a global search method based on population, which can evolve the population to the optimal solution through mutation crossover and selection. It has the advantages of fewer control parameters, fast convergence and strong robustness [24,25]. The DE algorithm has been comprehensively developed in recent decades, producing, for example, the DREAM algorithm. Luo et al. [26] studied the identification of the spatial variability of aquifer hydraulic conductivity based on the DREAM algorithm, providing a new idea for the study of spatial variability of aquifer parameters. Yang et al. [27] used the DREAM algorithm to analyze the factors affecting the uncertainty of groundwater numerical simulation. Zhang et al. [28] studied the probabilistic back analysis of soil parameters and displacement prediction of unsaturated slopes using Bayesian updating.

The problem of fluid–structure coupling is a key concern in engineering construction, and back analysis based on the fluid–structure coupling problem has been studied in recent years. Wu et al. [29] proposed a probabilistic back analysis method based on polynomial chaos expansion. Based on stochastic polynomial expansion, the probabilistic back analysis of fluid–structure coupling for an unsaturated soil slope was developed. Wang et al. [30] studied the inversion method of dams' seepage characteristics based on fluid–structure coupling. Based on the fluid–solid coupling theory and Bayesian theory, Zheng et al. [31] established a coupled probabilistic back analysis model for an unsaturated soil slope. A method of multi-objective probabilistic inverse analysis using time-varied data of displacement and pore water pressure was proposed based on Markov chain theory. Xu et al. [32] studied the coupled grouting reinforcement mechanism and displacement back analysis of mechanical parameters of surrounding rock.

In this paper, the GP algorithm and DE algorithm (GP-DE) are introduced in the parameter identification of fluid–solid coupling of the surrounding rock of a tunnel. Firstly, the numerical calculation model was established and solved using the fluid–solid coupling model. In orthogonal design analysis, the displacement of surrounding rock and pore water pressure distribution in different rock mass parameter combinations were obtained, and the learning samples needed for machine learning were established. Then, GP was used to obtain the nonlinear mapping relationship in the learning samples, and DE was used to optimize the critical parameters involved in this process. Furthermore, according to the established regression model and the measured displacement and pore water pressure in the research area, DE was used again to optimize the rock mass parameters and obtain the parameter feedback analysis results. In addition, in order to improve the mapping effect, the super-parameters of the GP model were optimized by the DE algorithm. Then, the trained GP model was integrated into the DE algorithm to identify the parameters of the tunnel-surrounding rock. Finally, the method was applied to the Chenjiadian tunnel in the city of Dalian, China. Through this method, the optimization of tunnel excavation

footage is realized, and the construction efficiency is effectively improved on the premise of ensuring safety.

## 2. The Parameter Identification Method of Fluid–Solid Coupling of Surrounding Rock Based on GP-DE

### 2.1. The Problem of Parameter Inversion of Fluid–Solid Coupling of Surrounding Rock

The identification of parameters is essentially an optimization problem. The optimization process can be expressed as:

$$\min E(x_1, x_2, \cdots, x_N) = \min(\frac{1}{m}\sum_{k=1}^{m}|Y_k^0 - Y_k|)$$
$$x_k^a \leq x_k \leq x_k^b \quad (k = 1, 2, \cdots, N) \tag{1}$$

where $E$ represents the mapping function between surrounding rock parameters and tunnel displacement, $Y_k^0$ is the field monitoring result of tunnel displacement and $Y_k$ is the tunnel displacement calculated through numerical simulation. $m$ is the number of observed values, $x_k$ is the surrounding rock parameter, $N$ is the number of parameters and $x_k^a$ and $x_k^b$ are the upper and lower limits.

### 2.2. The GP Algorithm

In the process of back analysis, the operation process of GP is as follows.

Assume $\mathbf{X} = [x_1, x_2, \ldots, x_n]$ is the $d \times n$ input matrix, and $\mathbf{y} = [y_1, y_2, \ldots, y_n]$ is the output vector, then the training dataset can be expressed as $\{\mathbf{X}, \mathbf{y}\}$; thus, the standard linear regression model with Gaussian white noise can be expressed as:

$$y_i = f(x_i) + \varepsilon \tag{2}$$

where $\varepsilon$ denotes an independent random variable, and $\varepsilon \sim N(0, \sigma_n^2)$, while $\sigma_n^2$ represents the variance.

The prior distribution of the observed target value $\mathbf{y}$ can be expressed as:

$$\mathbf{y} \sim N\left(0, \boldsymbol{C} + \sigma_n^2 \boldsymbol{I}\right) \tag{3}$$

where $C = C(\mathbf{X}, \mathbf{X})$ denotes a symmetric positive definite covariance matrix of the $n$th order.

For the test sample $(\mathbf{x}^*, \mathbf{y}^*)$, where $\mathbf{x}^* = (x_1^*, x_2^*, x_3^*, \ldots, x_n^*)$, $\mathbf{y}^* = (y_1^*, y_2^*, y_3^*, \ldots, y_n^*)$, the joint Gaussian prior distribution of $\mathbf{y}$ and $\mathbf{y}^*$ can be obtained and expressed as:

$$\begin{bmatrix} \mathbf{y} \\ \mathbf{y}^* \end{bmatrix} \sim N\left(0, \begin{bmatrix} C(\mathbf{X}, \mathbf{X}) + \sigma_n^2 \boldsymbol{I} & C(\mathbf{X}, \mathbf{x}^*) \\ C(\mathbf{x}^*, \mathbf{X}) & C(\mathbf{x}^*, \mathbf{x}^*) \end{bmatrix}\right) \tag{4}$$

where $C(\mathbf{X}, \mathbf{X})$ denotes an $n \times n$ symmetric positive definite covariance matrix, and $I$ represents the identity matrix. $C(\mathbf{X}, \mathbf{x}^*) = C(\mathbf{x}^*, \mathbf{X})^{\mathrm{T}}$ is an $n \times 1$ covariance matrix consisting of new input test points $\mathbf{x}^*$ and all input points; $\mathbf{C}(\mathbf{x}^*, \mathbf{x}^*)$ is the covariance matrix consisting of new input test points $\mathbf{x}^*$.

When the training set $D$ and the input value $\mathbf{x}^*$ of a test sample are known, the GP can use the posterior probability formula to calculate the output value $\mathbf{y}^*$ of the test sample, which can be expressed as:

$$\mathbf{y}^* | \mathbf{x}^*, D \sim N(u_{y^*}, \sigma_{y^*}^2) \tag{5}$$

$$u_{y^*} = C(\mathbf{x}^*, \mathbf{X})(C(\mathbf{X}, \mathbf{X}) + \sigma_n^2 \boldsymbol{I})^{-1} \boldsymbol{y} \tag{6}$$

where $u_{y^*}$ and $\sigma_{y^*}^2$ denote the expectation and variance of $\mathbf{y}^*$, respectively.

According to the Gaussian process, the covariance function is used to measure the degree of similarity between the learning sample and the prediction sample. In this case, the covariance function is similar to the kernel function of support vector machine, which plays an important role in Gaussian process machine learning methods. For the rest of the

calculation, the constructed covariance function can meet the requirements of symmetry and positive qualitative. In the early trial calculation process, it is determined to choose the square index covariance function, and its prediction effect is better. It is expressed as:

$$k_{se}(x_p, x_q) = \sigma_f^2 \exp\left(-\frac{1}{2J^2}\|x_p - x_q\|^2\right) + \sigma_n^2 \delta_{pq} \tag{7}$$

where $x_p$ and $x_q$ can represent the learning samples, prediction samples or combinations of learning and prediction samples depending on a particular situation; $J$ is the distance correlation between the two data points $x_p$ and $x_q$; $\sigma_f$ is the local correlation; $\sigma_n$ is the standard deviation of the noise; and lastly, $\delta_{pq}$ is a sign function. When $p = q$, then $\delta_{pq} = 0$; otherwise, $\delta_{pq} = 1$.

The GP-based surface should be trained by representative data samples before it can map the complex nonlinear relation between the jointed parameters and displacements. The data samples can be obtained by model tests, field tests, numerical simulation and other methods. In this study, the data samples were collected using the orthogonal design, uniform design and numerical simulation. In the GP training process, hyper-parameters $\sigma_f$ and $\sigma_n$ affect the GP training effect and prediction accuracy, so this process can be described as an optimization problem, which is expressed as:

$$\min E(\theta) = \min\left(\sum_{h=1}^{K} \frac{GP_h(\theta) - Y_h}{Y_h}\right), h = 1, 2, \ldots K \tag{8}$$

where $GP_h(\theta)$ and $Y_h$ denote the estimated output data of the tentative GP and the real output corresponding to the $h$th test sample. The test sample number is $h = 1, 2, \ldots, K$. $\theta = (\sigma_f, \sigma_n)$ represents the hyper-parametric vector.

### 2.3. The GP Optimized by DE

During the feedback analysis process, an intelligent optimization algorithm, DE, was used to optimize $\theta$ in this study. The basic operations of the algorithm include four steps:

(1) Generating initial population

Generate the initial search point; that is, generate the original population $P_G$:

$$P_G = \left\{\vec{x}_1, \cdots \vec{x}_i \cdots \vec{x}_{Np}\right\}, i = 1, \cdots, NP \tag{9}$$

where $G$ is evolutionary algebra and $NP$ is the population size, and its value does not change with evolution. Individual B is expressed as:

$$\vec{x}_i = (x_{i,1}, \cdots x_{i,j}, \cdots x_{i,n}), i = 1, \cdots, NP, j = 1, \cdots, n \tag{10}$$

The $j$th component $x_{i,j}$ of the individual $\vec{x}_i = (x_{i,1}, \cdots x_{i,j}, \cdots x_{i,n})$ in the initial population $P_0$ is randomly generated in the search space S, where S refers to the boundary constraint condition of the problem to be optimized:

$$x_{i,j} = Lbound_j + rand \times (Ubound_j - Lbound_j),$$
$$i = 1, \cdots, NP, j = 1, \cdots n \tag{11}$$

where $n$ represents the individual dimension, $Ubound_j$ and $Lbound_j$ represent the upper and lower limits of components, respectively, and rand represents the random number that follows uniform distribution within the range of [0, 1].

(2) Mutation operation

Perform the mutation operation. Two target individuals are taken as a group to generate variation vectors:

$$v_i = ax_1 + bx_2 \tag{12}$$

where $a$ and $b$ are randomly generated weight coefficients, and $a + b = 1$.

(3)   Crossover operation

Crossover operation is performed on the variation vector obtained in the previous step and its corresponding target vector, and then the test vector is obtained:

$$u_{i,j} = \begin{cases} v_{i,j}, & for\ j = \langle l \rangle_n, \langle l+1 \rangle_n, \cdots, \langle l+L+1 \rangle_n \\ x_{i,j}, & otherwise \end{cases} \tag{13}$$

where $i = 1..., NP, j = 1,...,n$ is the modulo taking function with modulo $n$; l is an integer, which is randomly selected and generated in the interval $[1, n]$. $L$ is the number of experimental vectors generated through crossover operations.

(4)   Selection operation

The fitness of the test vectors generated by crossover was evaluated and compared with the original vector, and the vectors with better fitness were reserved for entering the new iteration process.

### 2.4. The Parameters Identification Flowchart

For Equation (1), the Yk can be calculated by the GP model, and then it is expressed as Equation (14). Adopting DE, the parameters of rock mass can be identified.

$$\min E(x_1, x_2, \cdots, x_N) = \min(\frac{1}{m} \sum_{j=1}^{m} \left| GP(x_1, x_2, \cdots, x_N)_j^0 - Y_j \right|)$$
$$x_k^a \leq x_k \leq x_k^b \quad (k = 1, 2, \ldots, N) \tag{14}$$

The process of back analysis of surrounding rock parameters is shown in Figure 1. The specific algorithm is as follows:

(1)   Orthogonal samples are obtained by numerical simulation, and learning samples are established according to the samples.
(2)   GP is used to learn the rules of learning samples.
(3)   The DE method is used to generate the initial population.
(4)   The mapping established in step 2 is called to calculate the output variables corresponding to the initial population in step 3.
(5)   Compare the calculated results of the previous step with the field-measured results. Enter step 7 when it meets the fitness requirements; otherwise, enter step 6.
(6)   Perform the DE optimization operation described above to generate a new initial population, and return to step 4.
(7)   Obtain and record the population at this time, and this result is the target parameter of the required back analysis.

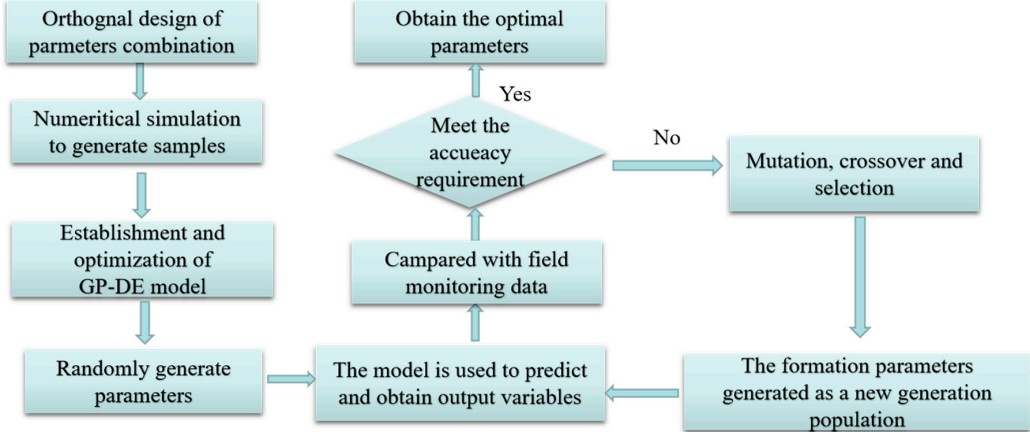

**Figure 1.** The flow diagram of the anti-analysis method which is based on the GP-DE algorithm.

## 3. Engineering Application

### 3.1. Engineering Overview

The Chenjiadian tunnel is 1500 m long, 10.5 m high and 12.7 m wide. From top to bottom, the tunnel geology contains a local surrounding rock fracture zone and abundant groundwater. The coupling action of water and rock and soil reduces the strength of the surrounding rock, which seriously affects the stability of the tunnel (Figure 2). Due to the complexity of geology and the limitations of exploration conditions, it is necessary to use an intelligent algorithm to determine the hydrogeological parameters.

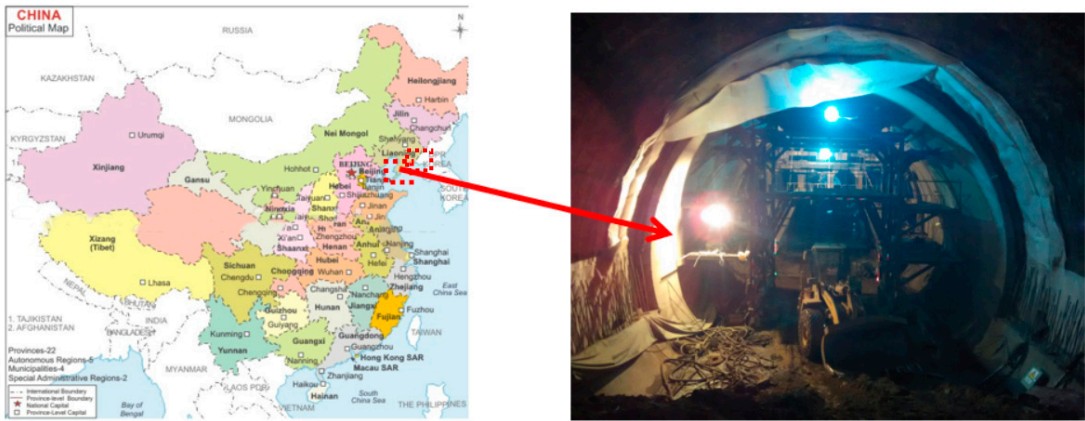

**Figure 2.** The location of the Chenjiadian tunnel.

### 3.2. The Principle of Fluid–Solid Coupling Modeling

FLAC3D software conducts fluid–structure coupling calculations based on the finite difference method. It defaults that the rock and soil mass are continuous media. The fluid seepage follows Darcy's law and satisfies Biot's equation, mainly including the following equations:

(1)  Equilibrium equation

For small deformation, the fluid particle equilibrium equation is:

$$-q_{i,i} + q_v = \frac{\partial \varsigma}{\partial t} \tag{15}$$

where $q_{i,i}$ is seepage velocity (m/s); $q_v$ is the volume fluid source intensity (s$^{-1}$); $\varsigma$ is the change in fluid volume per unit volume of porous media.

$$\frac{\partial \varsigma}{\partial t} = \frac{1}{M}\frac{\partial p}{\partial t} + \alpha\frac{\partial \varepsilon}{\partial t} - \beta\frac{\partial T}{\partial t} \tag{16}$$

where $M$ is the Biot modulus (N/m$^2$); $\alpha$ is the Biot coefficient; $\beta$ is the coefficient of thermal expansion ($°C^{-1}$), which considers liquid and solid particles. $p$ is the pore water pressure (Pa); $\varepsilon$ is the volume strain; $T$ is the temperature.

The momentum balance equation is:

$$\sigma_{ij,j} + \rho g_i = \rho\frac{dv_i}{dt} \tag{17}$$

$$\rho = (1-n)\rho_s + n\rho_w \tag{18}$$

Among them, $\rho$ is the bulk density (kg/m$^3$); $\rho_s$ and $\rho_w$ are the density of solid and liquid, respectively; $n$ is porosity; $g_i$ is the component of gravity acceleration (m/s$^2$); $v_i$ is the velocity component of the medium (m/s).

$$q_i = -k\left[p - \rho_f x_j g_j\right] \tag{19}$$

where $k$ is the permeability coefficient of the medium (m$^2$/(Pa·s)); $\rho_f$ is the fluid density (kg/m$^3$); $g_i$ is the component of gravity acceleration (m/s$^2$).

(2)   Constitutive equation

The volume strain and the pore pressure of the fluid interact with each other. The change in the strain makes the pore pressure readjust, and the change in the pore pressure also affects the occurrence of the strain. The descriptive equation is:

$$\Delta\sigma_{ij} + \alpha\Delta p\delta_{ij} = H_{ij}(\sigma_{ij}, \Delta\xi_{ij}) \tag{20}$$

Among them, $\Delta\sigma_{ij}$ is the stress increment; $\Delta p$ is the pore water pressure increment; $\delta_{ij}$ is the Kronecher factor. $H_{ij}$ is the given function. $\Delta\xi_{ij}$ is the total strain increment.

(3)   Compatibility equation

The relationship between strain rate and velocity gradient is:

$$\varepsilon_{ij} = \frac{1}{2}(v_{i,j} + v_{j,i}) \tag{21}$$

where $v$ is the velocity of a point in the medium (m/s).

(4)   Boundary condition

There are four types of boundary conditions in seepage calculation: (1) the given pore water pressure, (2) the given velocity vector outside the boundary normal direction, (3) the impervious boundary given by default in the program and (4) the pervious boundary. The form of pervious boundary is as follows:

$$q_n = h(p - p_e) \tag{22}$$

where $q_n$ is the velocity component in the normal direction outside the boundary, $h$ is the leakage coefficient (m$^3$/(N·s)) and $p_e$ is the pore water pressure at the seepage outlet.

(5)   Time scale

The fluid and mechanics processes are involved in the fluid–structure coupling calculation, and the time scales in these two states need to be considered. The characteristic time can generally characterize the size of the time scale. The characteristic time of the mechanical process is expressed as follows:

$$t_c^m = \sqrt{\frac{\rho}{K_u + 4/3G}}L_c \tag{23}$$

where $K_u$ is the undrained bulk modulus; $G$ is the shear modulus; $\rho$ is the density; $L_c$ is the feature length (the average size of the model). The characteristic time of the fluid diffusion process is defined as:

$$t_c^f = \frac{L_c^2}{c} \tag{24}$$

where $L_c$ is the characteristic length of seepage (the average size of seepage path in the model) and $c$ is the diffusion rate, defined as the ratio of permeability coefficient to water storage coefficient:

$$c = \frac{k}{S} \tag{25}$$

*3.3. Numerical Simulation Model*

FLAC3D software was used for the numerical simulation of excavation and support. The model included the embankment of the He-Da Expressway with a slope of 1:1.5 (Figure 3a). At the bottom of the model were X, Y and Z constraints, and around them were normal constraints. The model consisted of 67,336 nodes and 62,784 elements, and the

Mohr–Coulomb yield criterion was adopted. Shell and cable structural elements simulated the primary support and bolt, respectively. Groundwater was located 6 m above the tunnel roof arch.

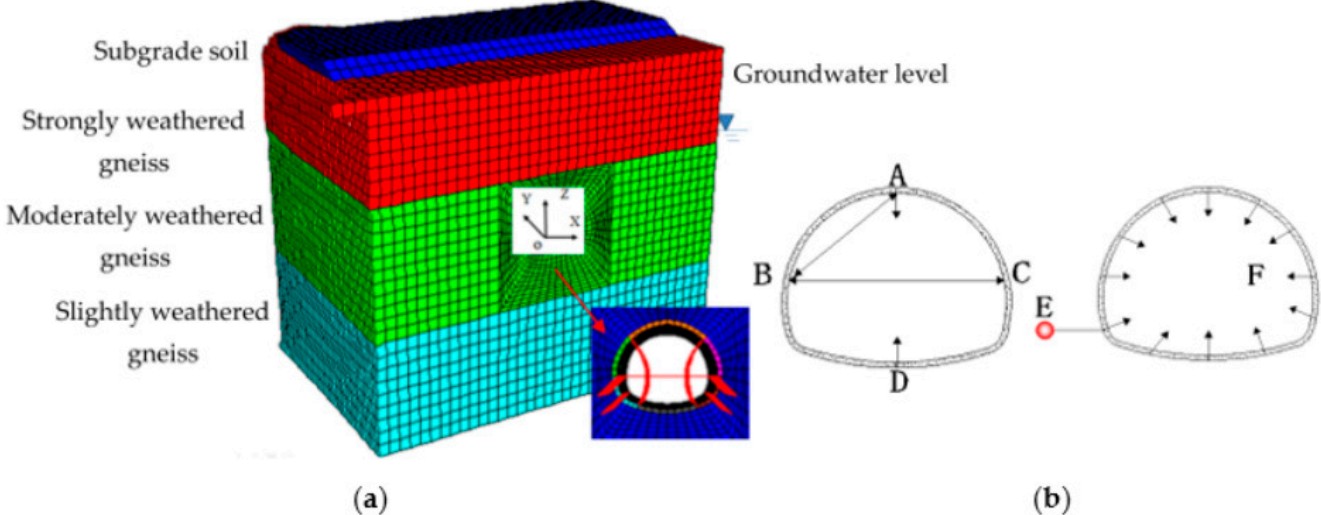

**Figure 3.** The numerical calculation model of the Chenjiadian tunnel. (**a**) Numerical model. (**b**) Distribution of monitoring points.

In the early stages of construction, the rock and soil masses within the construction range of the station are drilled and sampled. The geological parameters of the station are investigated and the stratigraphic parameters are obtained. However, there are some limitations in the geological surveys. The process of geological surveying is interfered with by many factors. The rock mass and overlying rock mass parameters of the station are essential references in the construction process. The main body of the tunnel in the study area is in moderately weathered gneiss, and the strongly weathered gneiss above the tunnel affects the stability of the tunnel to a certain extent. Therefore, the relevant parameters of moderately weathered gneiss and strongly weathered gneiss are mainly identified.

The elastic modulus (E1) and Poisson's (μ1) ratio of moderately weathered gneiss and the elastic modulus (E2) and Poisson's ratio (μ2) of strongly weathered gneiss were selected as the back analysis parameters. In addition, the permeability coefficients of moderately weathered gneiss (K1) and the permeability coefficients of strongly weathered gneiss (K2) that are difficult to measure were also added as the back analysis parameters. Other mechanical parameters are shown in Table 1. Input values were tunnel deformation value, pore water pressure and water inflow, and output values were surrounding rock parameters.

The arch crown settlement AZ, arch bottom uplift DZ, arch waist convergence BC and the relative displacement AB of measuring points A and B were taken as the displacement monitoring values (Figure 3b). The pore water pressure P at point E and the unit seepage volume F of the tunnel were taken as the seepage monitoring values. Among them, point E was 1 m away from the arch foot of the tunnel, the pore water pressure P was measured by the pore water pressure gauge and the unit seepage F of the tunnel was calculated by dividing the sum of the seepage of all outlet points in a particular mileage section of the tunnel by the length.

According to relevant specifications for tunnel engineering design and geological survey data, the value ranges of six parameters were as follows: E1 is 2.85 GPa~6.53 GPa; μ1 is 0.21~0.41; E2 is 1.81 GPa~3.49 GPa; μ2 is 0.25~0.45; K1 is 0.196 m/d~0.372 m/d; K2 is 0.4 m/d~0.78 m/d. The orthogonal design scheme and uniform design scheme were established through these six parameters for numerical calculation. The calculation results are shown in Table 2 (training samples) and Table 3 (test samples).

**Table 1.** The parameters of calculation.

|  | Elastic Modulus/GPa | Poisson's Ratio | Cohesion /kPa | Internal Friction Angle/° | Permeability Coefficient/(m/d) |
|---|---|---|---|---|---|
| Slightly weathered gneiss | 4.19 | 0.26 | 27 | 38 | 0.025 |
| Moderately weathered gneiss | — — | — — | 21 | 42 | — — |
| Strongly weathered gneiss | — — | — — | 15 | 46 | — — |
| Subgrade soil | 0.15 | 0.35 | 23 | 19 | 0.843 |
| Primary support | 25 | 0.18 | 20,000 | 34 | $6.3 \times 10^{-4}$ |
| Bolt | 200 | — — | — — | 25 | — — |
| Middle wall | 25 | 0.18 | 20,000 | 34 | — — |

**Table 2.** Tunnel-surrounding rock parameters' orthogonal scheme and the calculation results.

|  | E1 (GPa) | μ1 | E2 (GPa) | μ2 | K1 (m/d) | K2 (m/d) | AZ (mm) | AB (mm) | BC (mm) | DZ (mm) | P ($10^5$) (Pa) | F(m³/m × d) |
|---|---|---|---|---|---|---|---|---|---|---|---|---|
| 1 | 2.85 | 0.21 | 1.81 | 0.25 | 0.196 | 0.4 | 5.961 | 3.64 | 0.355 | 1.313 | 1.337 | 9.07 |
| 2 | 3.52 | 0.26 | 1.81 | 0.3 | 0.24 | 0.495 | 5.445 | 3.481 | 0.291 | 1.186 | 1.264 | 8.866 |
| 3 | 4.19 | 0.31 | 1.81 | 0.35 | 0.284 | 0.59 | 5.002 | 3.342 | 0.763 | 1.097 | 1.16 | 5.34 |
| 4 | 5.86 | 0.36 | 1.81 | 0.4 | 0.328 | 0.685 | 4.513 | 3.224 | 1.425 | 0.982 | 1.026 | 3.1 |
| 5 | 6.53 | 0.41 | 1.81 | 0.45 | 0.372 | 0.78 | 4.12 | 3.175 | 2.149 | 1.015 | 0.84 | 3.04 |
| 6 | 4.19 | 0.21 | 2.23 | 0.3 | 0.328 | 0.78 | 5.193 | 3.471 | 0.165 | 1.496 | 1.348 | 4.31 |
| 7 | 5.86 | 0.26 | 2.23 | 0.35 | 0.372 | 0.4 | 4.717 | 3.288 | 0.002 | 0.921 | 1.311 | 3.74 |
| 8 | 6.53 | 0.31 | 2.23 | 0.4 | 0.196 | 0.495 | 4.431 | 3.202 | 0.753 | 0.883 | 1.162 | 3.52 |
| 9 | 2.85 | 0.36 | 2.23 | 0.45 | 0.24 | 0.59 | 4.362 | 3.234 | 1.035 | 1.656 | 1.045 | 3.33 |
| 10 | 3.52 | 0.41 | 2.23 | 0.25 | 0.284 | 0.685 | 4.773 | 3.167 | 1.696 | 0.706 | 0.721 | 6.34 |
| 11 | 6.53 | 0.21 | 2.65 | 0.35 | 0.24 | 0.685 | 4.598 | 3.306 | 0.018 | 0.9 | 1.358 | 3.23 |
| 12 | 2.85 | 0.26 | 2.65 | 0.4 | 0.284 | 0.78 | 4.819 | 3.367 | 0.012 | 1.494 | 1.231 | 3.38 |
| 13 | 3.52 | 0.31 | 2.65 | 0.45 | 0.328 | 0.4 | 4.283 | 3.209 | 0.511 | 1.414 | 1.217 | 3.45 |
| 14 | 4.19 | 0.36 | 2.65 | 0.25 | 0.372 | 0.495 | 4.663 | 3.135 | 0.941 | 0.66 | 0.882 | 6.4 |
| 15 | 5.86 | 0.41 | 2.65 | 0.3 | 0.196 | 0.59 | 4.205 | 3.049 | 1.57 | 0.613 | 0.72 | 3.79 |
| 16 | 3.52 | 0.21 | 3.07 | 0.4 | 0.372 | 0.59 | 4.639 | 3.343 | 0.233 | 1.318 | 1.355 | 3.37 |
| 17 | 4.19 | 0.26 | 3.07 | 0.45 | 0.196 | 0.685 | 4.24 | 3.232 | 0.151 | 1.287 | 1.283 | 3.18 |
| 18 | 5.86 | 0.31 | 3.07 | 0.25 | 0.24 | 0.78 | 4.462 | 3.143 | 0.48 | 0.644 | 0.977 | 3.35 |
| 19 | 6.53 | 0.36 | 3.07 | 0.3 | 0.284 | 0.4 | 4.184 | 3.039 | 0.927 | 0.572 | 0.895 | 3.89 |
| 20 | 2.85 | 0.41 | 3.07 | 0.35 | 0.328 | 0.495 | 4.469 | 3.097 | 1.261 | 0.959 | 0.749 | 7.38 |
| 21 | 5.86 | 0.21 | 3.49 | 0.45 | 0.284 | 0.495 | 4.123 | 3.2 | 0.03 | 1.028 | 1.398 | 4.5 |
| 22 | 6.53 | 0.26 | 3.49 | 0.25 | 0.328 | 0.59 | 4.38 | 3.156 | 0.151 | 0.642 | 1.114 | 3.36 |
| 23 | 2.85 | 0.31 | 3.49 | 0.3 | 0.372 | 0.685 | 4.857 | 3.226 | 0.227 | 0.89 | 0.95 | 6.56 |
| 24 | 3.52 | 0.36 | 3.49 | 0.35 | 0.196 | 0.78 | 4.39 | 3.111 | 0.692 | 0.88 | 0.829 | 3.86 |
| 25 | 4.19 | 0.41 | 3.49 | 0.4 | 0.24 | 0.4 | 3.959 | 3.004 | 1.214 | 0.908 | 0.761 | 6.17 |

**Table 3.** Uniform parameter test methods and the results of numerical calculation.

|  | E1 (GPa) | μ1 | E2 (GPa) | μ2 | K1 (m/d) | K2 (m/d) | AZ (mm) | AB (mm) | BC (mm) | DZ (mm) | P ($10^5$) (Pa) | F (m³/m × d) |
|---|---|---|---|---|---|---|---|---|---|---|---|---|
| 1 | 4.19 | 0.26 | 1.81 | 0.035 | 0.372 | 0.875 | 5.053 | 3.352 | 1.175 | 1.164 | 1.090 | 6.335 |
| 2 | 5.86 | 0.36 | 2.23 | 0.20 | 0.284 | 0.780 | 4.374 | 3.136 | 0.447 | 0.797 | 0.861 | 4.291 |
| 3 | 3.52 | 0.46 | 2.65 | 0.40 | 0.196 | 0.685 | 4.049 | 3.032 | 0.698 | 0.621 | 0.750 | 3.310 |
| 4 | 6.53 | 0.21 | 3.07 | 0.25 | 0.416 | 0.590 | 4.503 | 3.177 | 0.586 | 0.867 | 0.904 | 4.680 |
| 5 | 2.85 | 0.31 | 3.49 | 0.45 | 0.328 | 0.495 | 4.249 | 3.096 | 0.313 | 0.729 | 0.818 | 3.915 |

### 3.4. Parameters Identification Results

During the GP-DE feedback analysis process, the population size NP was 100, the variation factor F was 0.7, the cross factor CR was 0.9, the maximum evolutionary algebra was 200 and SE was selected as the kernel function. After tunnel excavation, the measured values of AZ, DZ, BC, AB, P and F were 8 mm, 3.17 mm, 3.02 mm, 6.31 mm, 0.122 MPa and 3.77 m3/m × d, respectively. The optimal parameters obtained by back analysis were E1 = 2.83 GPa, μ1 = 0.33, E2 = 1.24 GPa, μ2 = 0.36, K1 = 0.285 m/d and K2 = 0.658 m/d.

According to Table 4, compared with the numerical simulation results, the maximum relative error of the back analysis is 9.40%, which meets the requirements of engineering construction. This back analysis method can be used for surrounding rock parameter prediction.

**Table 4.** Parameters of surrounding rock of uniform testing scheme analysis results.

| | Back Analysis Result | | | | | | Relative Error | | | | | |
| | E1 (GPa) | μ1 | E2 (GPa) | μ2 | K1 (m/d) | K2 (m/d) | E1 (%) | μ1 (%) | E2 (%) | μ2 (%) | K1 (%) | K2 (%) |
| --- | --- | --- | --- | --- | --- | --- | --- | --- | --- | --- | --- | --- |
| 1 | 3.85 | 0.29 | 1.81 | 0.33 | 0.36 | 0.82 | 0.00 | −9.40 | 8.74 | 6.81 | 4.49 | 7.24 |
| 2 | 6.96 | 0.35 | 2.39 | 0.22 | 0.26 | 0.78 | −6.69 | 3.79 | 3.39 | −8.19 | 9.15 | 0.00 |
| 3 | 3.37 | 0.47 | 2.49 | 0.37 | 0.20 | 0.73 | 6.43 | −1.72 | 4.54 | 7.50 | 0.00 | −6.16 |
| 4 | 6.19 | 0.23 | 3.26 | 0.27 | 0.45 | 0.58 | −5.80 | −8.41 | 5.56 | −8.45 | −7.01 | 2.02 |
| 5 | 3.07 | 0.34 | 3.49 | 0.41 | 0.33 | 0.52 | 0.00 | −7.98 | −7.14 | 9.22 | −1.60 | −4.81 |
| 6 | 6.43 | 0.39 | 3.74 | 0.28 | 0.25 | 0.38 | 4.55 | 6.33 | −8.85 | 7.14 | −4.57 | 5.26 |

Figure 4 shows the fitness values in the iterative process. It can be seen from the figure that with the increase in evolutionary algebra, the distribution of solution vector in space tends to converge. The iterative process of parameter acquisition is shown in Figure 5. In the initial stage of iteration, the fluctuation range of parameters is extensive. When the number of iterations reaches 45, the obtained parameters no longer fluctuate and the optimal solution is generated.

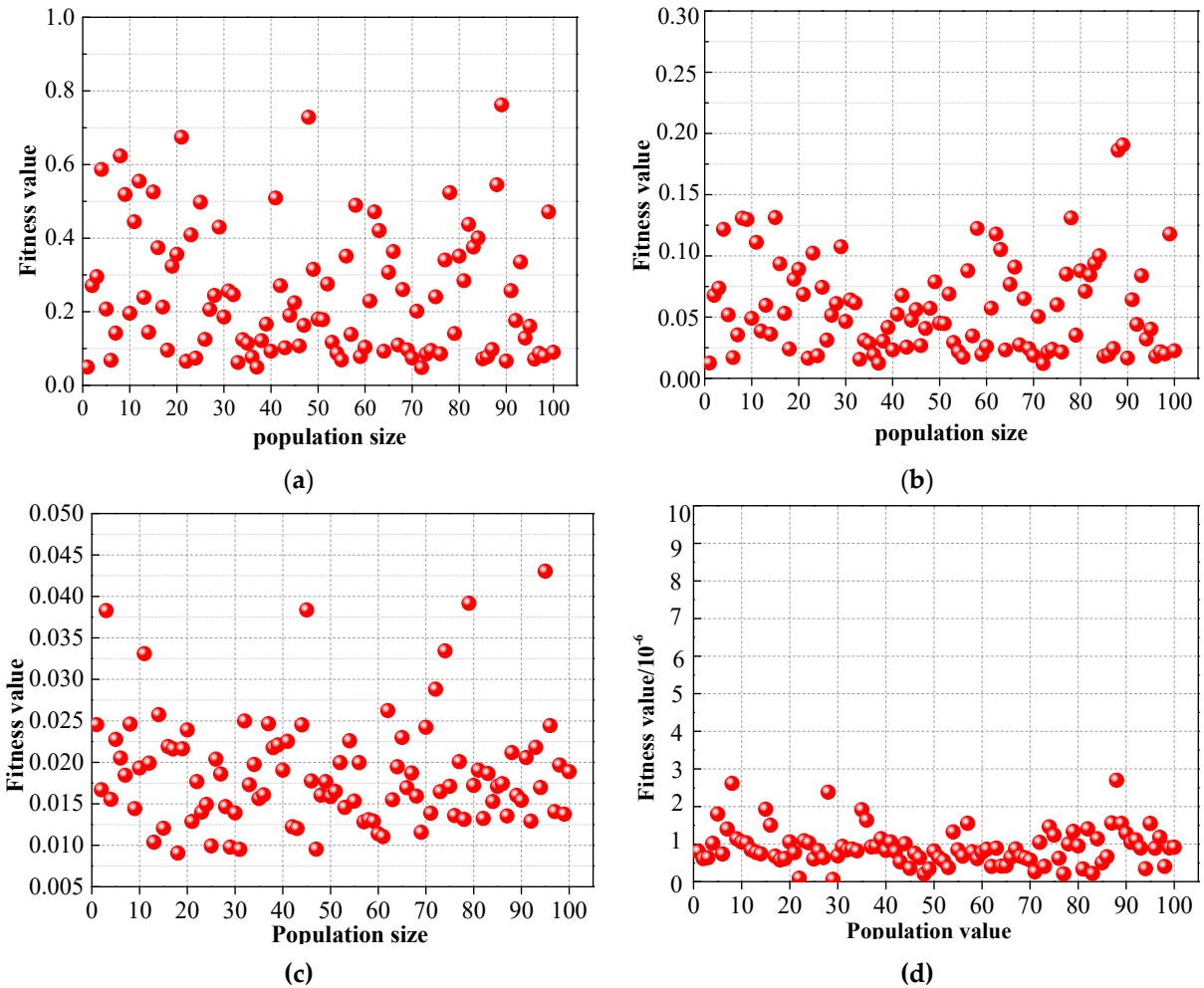

**Figure 4.** Fitness values with different evolutions. (**a**) 1st evolution. (**b**) 15th evolution. (**c**) 30th evolution. (**d**) 45th evolution.

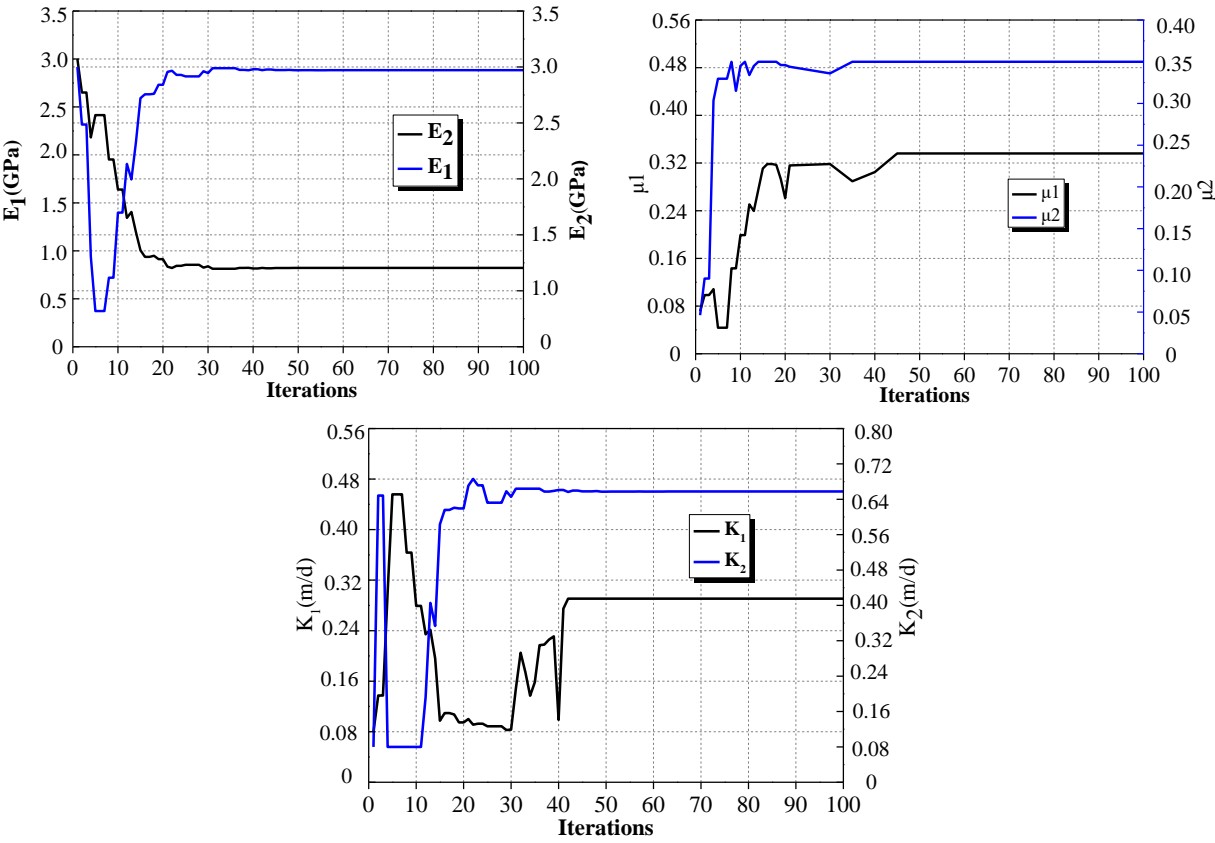

**Figure 5.** Variation in the recognized parameters.

### 3.5. Analysis of Tunnel Excavation Footage Based on Back Analysis Results

Based on the back analysis results, different amounts of excavation footage of the tunnel were selected to analyze their impact on the tunnel. The optimal amounts of excavation footage under four working conditions were selected by analyzing the distribution of the plastic zone. The distribution of the plastic zone under four cyclic excavation footage conditions is shown in Figure 6.

It can be seen from Figure 6 that the plastic zones in the four cyclic excavation footage conditions are distributed differently. Due to the reinforcement effect of the advanced small pipe, the plastic zone of the arch crown is reduced, and the plastic zone is mainly distributed on both sides of the arch foot and arch shoulder. With the increase in excavation footage, the pressure release from surrounding rock also increases, so the area of the plastic zone increases obviously. The area of the plastic zone is the largest under the condition of 2.5 m excavation footage. Considering the actual situation of the project, the excavation footage of 1.5 m should be selected for excavation. At the same time, grouting reinforcement should be strengthened on both sides of the arch foot and the arch shoulder of the tunnel to ensure construction safety.

According to the analysis results, the excavation footage of the construction site was determined to be 1.5 m. Figure 7 shows the site construction condition of the tunnel when it is constructed according to the excavation footage of 1.5 m. In the construction process, the surrounding rock of the tunnel is relatively stable and the construction environment is safe. The monitoring data of the arch settlement change obviously in the early stage of excavation, and gradually tend to be stable in the later stage. The arch settlement value is always within the monitoring control range in the monitoring process. In the construction process, the original 1.0 m excavation footage is adjusted to 1.5 m, which effectively improves the construction efficiency.

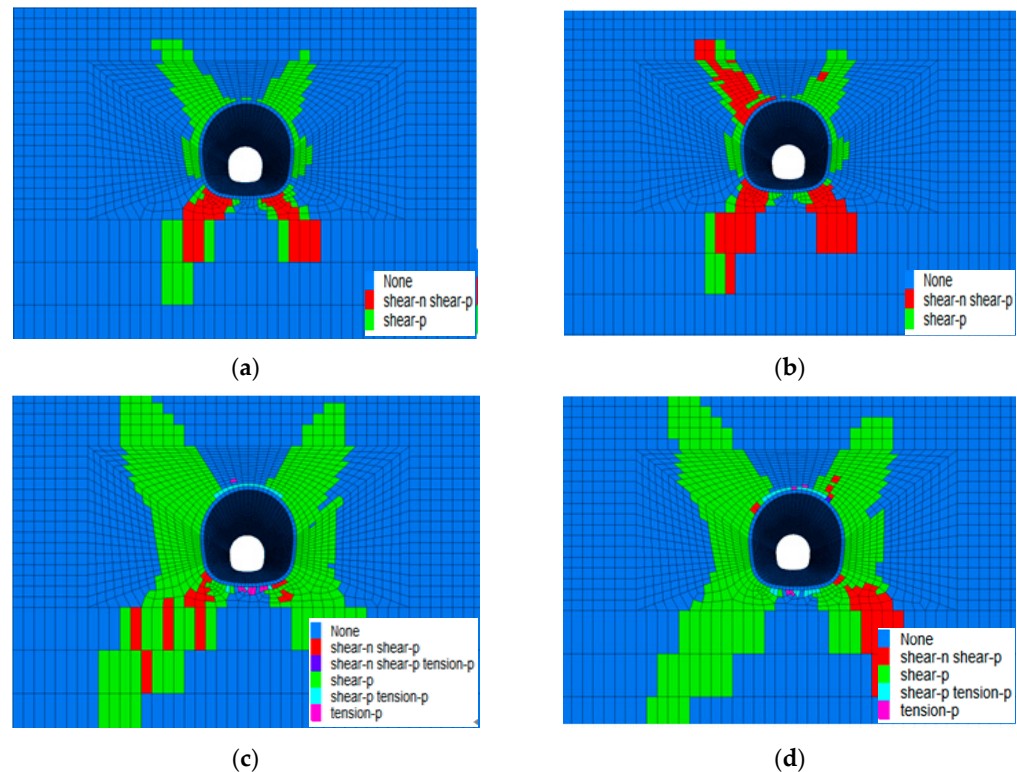

**Figure 6.** Plastic differentiation layout under various working conditions. (**a**) 1 m. (**b**) 1.5 m. (**c**) 2.0 m. (**d**) 2.5 m.

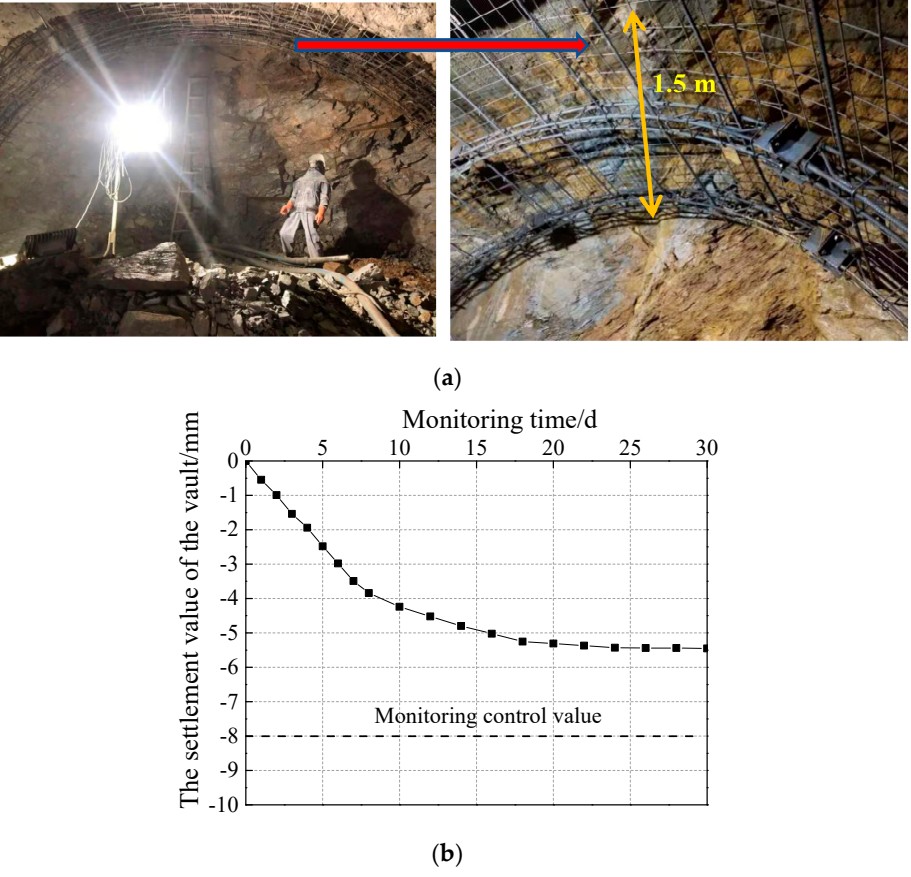

**Figure 7.** Optimization of tunneling excavation footage. (**a**) Optimized results of tunnel excavation at construction site. (**b**) Field monitoring data of vault settlement.

Figure 8 shows the results of vault settlement under different working conditions. In the simulation process, without considering the fluid–structure coupling, the arch settlement result caused by tunnel excavation is 4.26 mm, and when considering the fluid–structure coupling, the simulation result shows that the arch settlement is 5.32 mm. In the actual construction process, the maximum settlement of the vault is 5.45 mm. Therefore, when tunnel construction is carried out in water-rich areas, the numerical simulation results are closer to the actual on-site construction conditions when considering the fluid–structure coupling. Thus, the impact of groundwater on construction cannot be ignored. In the actual construction process, the corresponding water stop measures should be taken to ensure the safety of construction.

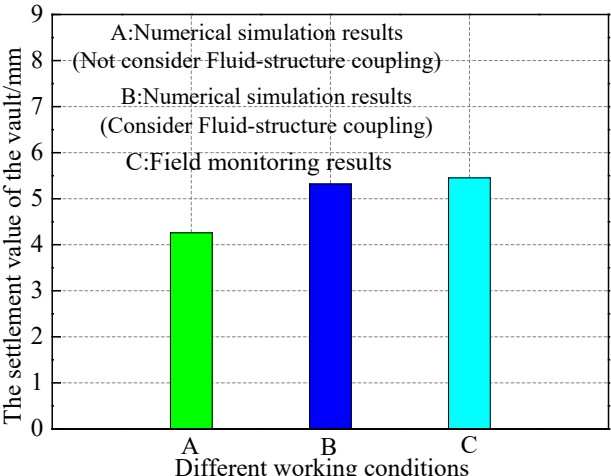

**Figure 8.** The vault settlement under different working conditions.

## 4. Discussion

### 4.1. The Influence of GP Parameters on the Results of Back Analysis

$\sigma_f$ and $\sigma_n$ are important super-parameters of the GP model. Figure 9 shows the prediction accuracy under different parameters. When $\ln \sigma_f = 3.56$ and $\ln \sigma_n = 8.72$, the relative error is 3.56%; thus, the accuracy of prediction is affected by the parameter selection. Therefore, in the back analysis of parameters, choosing the appropriate parameters for GP is important.

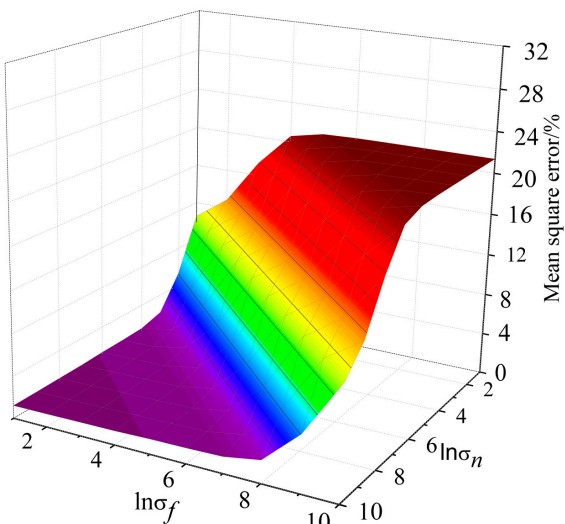

**Figure 9.** The influence of GP parameters on the prediction accuracy.

### 4.2. *The Influence of DE Parameters*

DE is more complex and involves many influencing factors in the GP-DE algorithm. F, CR, N and other difference strategies perhaps have an impact on the convergence speed. The DE/Best/1 difference strategy and NP = 100 were selected, with F = 0.6 and CR between 0.5 and 0.9. There was a difference in convergence speed in the process of optimization. When CR = 0.9, the number of iterative steps required to achieve convergence is the lowest. Selecting CR as 0.9 and F as 0.5~0.9, the convergence rate is the fastest when F = 0.7. It is shown that the appropriate initial parameters can improve the convergence speed. CR = 0.9 and F = 0.7 were selected for this study (Figure 10).

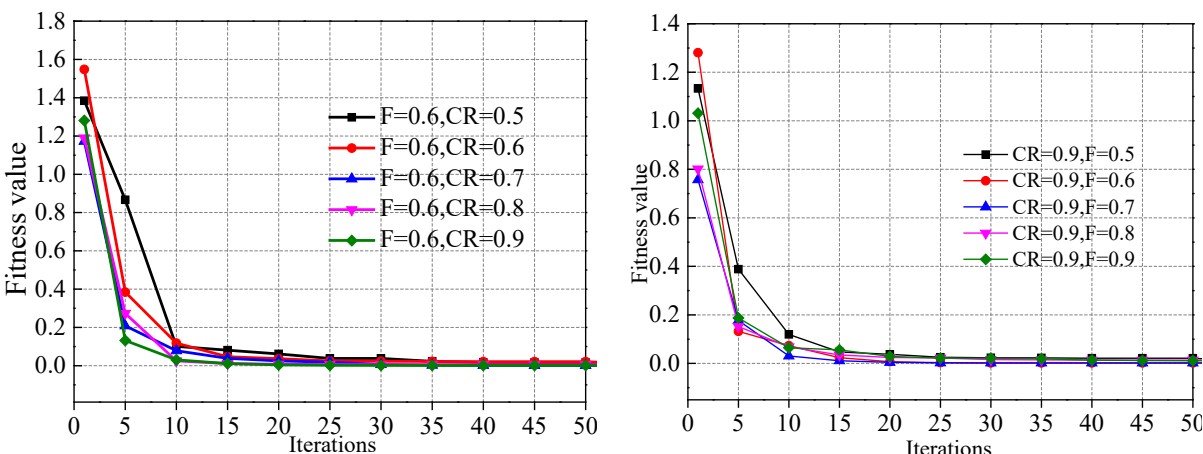

**Figure 10.** Iteration curve.

The DE/Best/1 difference strategy was selected, with CR = 0.9 and F = 0.7, and NP changed. As seen from Figure 11, when the population size reaches 100, the precision of parameter optimization no longer changes significantly with the increase in population.

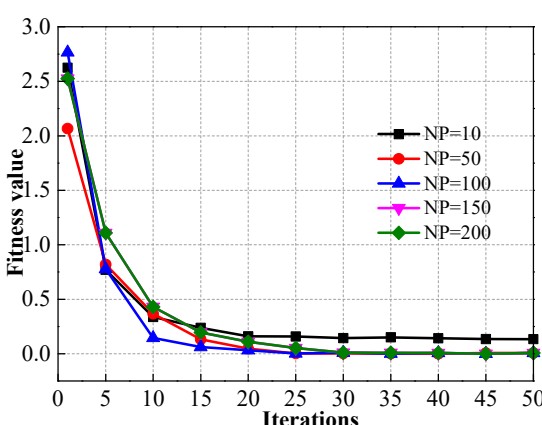

**Figure 11.** Iteration curve of different populations.

In the DE algorithm, there are various difference strategies, shown in Equation (26). Selecting F = 0.7, CR = 0.9 and NP = 100, the different strategies are compared. It can be seen from Figure 12, compared with other strategies, that DE/Best/1 is the best strategy for optimization.

$$
\begin{cases}
DE/rand/1 : v_{i,g} = x_{r1,g} + F\left(x_{r2,g} - x_{r3,g}\right) \\
DE/best/1 : v_{i,g} = x_{best,g} + F\left(x_{r2,g} - x_{r3,g}\right) \\
DE/rand/2 : v_{i,g} = x_{r1,g} + F\left(x_{r2,g} - x_{r3,g} + x_{r4,g} - x_{r5,g}\right) \\
DE/best/2 : v_{i,g} = x_{best,g} + F\left(x_{r1,g} - x_{r2,g} + x_{r3,g} - x_{r4,g}\right) \\
DE/rand - to - best/2 : v_{i,g} = x_{r1,g} + F\left(x_{best,g} - x_{r2,g} + x_{r3,g} - x_{r4,g}\right)
\end{cases}
\tag{26}
$$

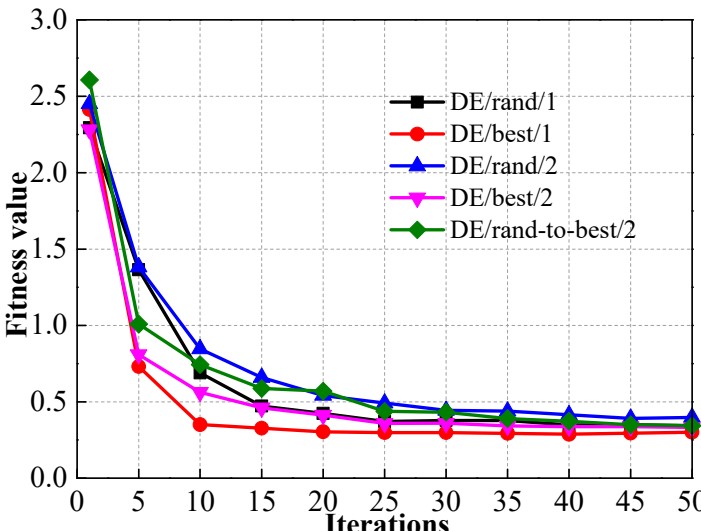

**Figure 12.** Various difference strategies.

## 5. Conclusions

Through back analysis of the surrounding rock parameters of the Chenjiadian tunnel and with the numerical calculation of the fluid–structure interaction, the following conclusions are obtained:

1. To realize parameter feedback optimization of tunnel construction in water-rich areas, a feedback analysis method of tunnel parameters under fluid–solid coupling conditions based on GP and DE was established based on an intelligent optimization algorithm.
2. Choosing the appropriate parameters of GP by DE is important to improve the accuracy of the back analysis results. The variation parameters of DE have an impact on the convergence speed. CR = 0.9, F = 0.7, N = 100 and the difference strategy DE/Best/1 were selected for this study.
3. The optimal hydrogeological parameters of the surrounding rock were obtained by a back analysis algorithm based on GP-DE. The optimal parameters from back analysis are E1 = 2.83 GPa, μ1 = 0.33, E2 = 1.24 GPa, μ2 = 0.36, K1 = 0.285 m/d and K2 = 0.658 m/d, providing an effective method for obtaining the surrounding rock parameters of similar projects.
4. Based on the back analysis results, different amounts of excavation footage of the tunnel were selected to analyze their impact on the tunnel. The optimal excavation footage under four working conditions was selected by analyzing the distribution of the plastic zone.

**Author Contributions:** Conceptualization, T.Z., T.J. and X.G.; methodology, T.Z., T.J. and A.J.; software, X.G., T.J. and A.J.; formal analysis, T.Z. and A.J.; resources, T.J. and A.J.; data curation, T.Z., T.J. and X.G.; writing—original draft preparation, T.Z., X.G. and A.J. All authors have read and agreed to the published version of the manuscript.

**Funding:** This work was supported by the National Natural Science Foundation of China (no. 52078093) and the Cultivation Program for the Excellent Doctoral Dissertation of Dalian Maritime University (no. 2022YBPY009).

**Institutional Review Board Statement:** Not applicable.

**Informed Consent Statement:** Informed consent was obtained from all subjects involved in the study.

**Conflicts of Interest:** The authors declare that they have no conflict of interest regarding the publication of this article.

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
