# Peer review of "Intelligent Feedback Analysis of Fluid–Solid Coupling of Surrounding Rock of Tunnel in Water-Rich Areas"

_applsci, doi:10.3390/app13031479_

Round 1

Reviewer 1 Report

This paper proposed an algorithm to estimate the rock parameters based on the fluid-solid coupling model of the tunnel. The topic is very interesting. However, there are several severe issues that I am concerned about, and based on this, I think this manuscript cannot meet the requirement of the journal.

1. Parameter estimation with a machine learning-based surrogate is quite well designed by previous studies and has not been improved in this study, especially for the interest of tunneling engineering. The contribution and novelty of this study are limited. 

2. This paper looks like a study of engineering examples, but in fact, it still is a hypothetical case. The field data is not involved. It does not contain substantial innovations in physics-based, physics-informed techniques or those that could offer new physical insights.

3. The written quality is poor both in the English language and in the scientific organization. How to verify the proposed method is not well designed. Many scientific details like the fluid-solid coupling modeling are missing. Although the language is correct in grammar, it lacks concise and clarity. The work reads like a translation of Chinese work.

Reviewer 2 Report

Dear Authors,

I read your manuscript very carefully.

I have some comments.

Figures quality should be improved.

The manuscript could be more understandable.

The parameters given in the tables should be illustrative.

The rock mass parameters using in the analysis are taken from the site reports. These parameters should be explained how it is determined.

Reviewer 3 Report

The following suggestions should be incorporated before the final acceptance of this paper which are as follows:

1. "1 Introduction": In the literature review, the author should also introduce the main conclusions of the previous works.

2. The quality of Figure 1 in the manuscript needs to be further improved.

3. The position of marks in Figure 2 needs to be further clarified.

4. The advantages of GP algorithm and DE algorithm selected in this paper need to be supplemented in this paper.

5. This paper needs to explain how to set the boundary conditions of 3D numerical model.

6. How to select the parameters of other strata needs to be supplemented in the paper.

7.Space should be set between units and values.

8.The language should be double checked or polished, there are some grammar mistakes and some descriptions are not quite suitable.

9.The conclusions is suggested to refine the main innovations and research results.

Round 2

Reviewer 1 Report

I feel the sincerity of the authors to revise this article. The structure of this manuscript looks good in this revised version. However, the low quality of the presentation really astonishes me, even after being polished by a professional academic editing agency. The manuscript still has many typos (e.g., line 16 gaussian, line 104 Where, line 90 regression, line 344 hyperparameters) and inappropriate symbols that do not conform to conventions (e.g. upright boldface for vectors ). Since the editor decided to provide the authors with a chance for revision, let me speak more to further improve the quality of this manuscript. 

1. Introduction:

1) The literature review is not in-depth. As the authors stated that the GP is a statistical method under the Bayesian framework, the scope of the literature review can be extended to Bayesian back analysis in geotechnical engineering.

2) One significant contribution of this study is a back analysis based on fluid-solid coupling. Therefore, recent literature on back analysis based on the coupling model should be cited.

3) DE algorithm has been comprehensively developed in recent decades such as the DREAM algorithm. The DREAM algorithm has been widely used recently to solve the geotechnical back analysis. It should be mentioned. This is also the reason why I judge the article lacks originality. 

2. Theory:

1) The theory part is horrible. Bear in mind that every symbol should be explained in texting and tell the readers how to acquire it. Some references are necessary. For example, the core of GP is Eq. 4. How to estimate it? What is E in Eq.1?

2) The authors should summarize the assumptions here. For example, why don't use the least squared objective function in Eq.1? Why choose square exponential covariance?

3. Results:

1) Since it is a fluid-solid coupling, why do not the authors use field data of pore water pressure for back analysis? If only settlement data is used, how to prove the merits of coupling data?

2) The engineers care about insights into engineering instead of the influence of hyperparameters, i.e., effects of monitoring duration or different types of data. The implications in engineering practice should be provided. 
